# Trauma-Informed Care: A Transcendental Phenomenology of the Experiences of International Faculty during the Delta and Omicron Variant Outbreaks in East China

**DOI:** 10.3390/ijerph191711057

**Published:** 2022-09-03

**Authors:** Benjamin H. Nam, Alexander S. English

**Affiliations:** 1School of Education, Shanghai International Studies University, Shanghai 201613, China; 2Department of Psychology and Behavioral Sciences, Zhejiang University, Hangzhou 310058, China

**Keywords:** mental health, trauma, international faculty, coping mechanisms, COVID-19

## Abstract

This transcendental phenomenological study explored psychologically traumatic incidents and risk factors among international faculty members (IFMs) who experienced long-term lockdowns during the Delta and Omicron outbreak periods in East China. Based on empirical voices from 18 IFMs in Shanghai, Hangzhou, and Nanjing, this study used trauma-informed care as its primary theoretical lens to examine potential traumatic incidents and risk factors. Findings showed that participants had neuroses about the omen of lockdowns and felt exhausted and frustrated about persistent polymerase chain reaction (PCR) tests. They also experienced or witnessed burnout and dropout due to leisure constraints. Most notably, participants had concerns about families and friends during the series of lockdowns, entailing extreme stress due to separation, illness, loss, and grief. Overall, this study provides practical implications for counseling practices about social and cultural considerations and systemic barriers that impact clients’ well-being.

## 1. Introduction

At the end of 2019, an unknown virus spread through Wuhan, the capital city of Hubei Province in China. The explosive chain of lethal transmission became severe in early 2020. The Chinese government initially sealed the city, the so-called “Wuhan’s 76-day lockdown” (23 January–8 April 2020). All residents were isolated in their own residences under the government’s social control in the early stage of the COVID-19 pandemic. During this specific timeframe, many residents in the city suffered long-term psychological trauma. They had to deal with stigmatization and anxiety about their physical and emotional well-being, as the city had the highest number of confirmed cases (67,733 out of 80,955) in China. In the meantime, the government implemented a nationwide lockdown policy [1,2,3].

Accordingly, diverse social functions collapsed from late January to the end of March; public transportation was interrupted, and local businesses were closed. Higher education institutions adopted online teaching, and primary and secondary education systems were temporarily closed [4]. The notions of lockdown, social distancing, and isolation due to the pandemic, which have influenced social and behavioral changes regarding mask use regardless of geographical location, socio-economic status, gender, and generation, were unknown and psychologically traumatic for numerous citizens [5,6].

While a growing body of evidence has shed light on mental health and social and behavioral changes among Chinese citizens in the early days of the COVID-19 pandemic, there has been relatively little research regarding mental health issues among foreigners, despite the fact that some recent studies investigated the pandemic experiences of international students and faculty during this timeframe and highlighted them as socially vulnerable populations without the emotional and social support of their families and friends [4,7,8,9]. Although China normalized the level of the public health crisis after the spring semester of 2020 (March–July), many international students who had stayed in their native countries could not return to China due to the ‘Zero-COVID’ policy that strictly reduced cross-border traffic [10,11]. Yet, numerous international faculty members were required to stay in China to sustain their professional and academic careers in teaching and research [10,11].

Despite the Delta variant outbreak during the summer through fall of 2021, Chinese citizens engaged in collective behavior to reduce confirmed cases, though residents in some key economic or metropolitan cities experienced a series of short-term (48 h) and long-term (14 days or more), city-wide lockdowns (e.g., Hangzhou, Nanjing, Shenzhen, Xian, and so on). Nonetheless, the currently ongoing (as of August 2022) Omicron outbreak, which started in the late fall of 2021, has caused the pandemic to enter a third year [8]. Hence, we argue that the Chinese national public health policy entails a raft of tightened restrictions and a series of lockdowns. This may have led many foreigners to raise critical questions about the sustainability of such an approach and to face emotional challenges while stuck in lockdown and forced to persistently undergo polymerase chain reaction (PCR) tests in the surrounding areas.

A recent study regarding the major Shanghai lockdown period (2 March–1 June 2022) showed that over 160,000 expats were isolated at their homes. This study observed a total of 1558 WeChat accounts and identified that numerous foreigners had emotional challenges due to frequent policy changes and drastic lifestyle shifts [8]. Despite the in-depth understanding of the lockdown experiences of expats, the previous scholarship was limited to describing vivid traumatic experiences of international faculty, while depicting frustration, anger, and stress. Notably, scholars have paid scant attention to the psychologically traumatic experiences of international faculty during the Delta variant outbreak.

In this paper, we explored the psychologically traumatic experiences of international faculty members (IFM) amid long-term lockdowns during the Delta and Omicron variant outbreaks in Shanghai, Hangzhou, and Nanjing, located in East China along the Yangtze River Delta region, which is a global financial hub. We aim to determine factors causing psychologically traumatic symptoms and analyze the development of coping mechanisms to deal with mental crises. To this end, we adopted the concepts of trauma-informed care (TIC) and transcendental phenomenology as our primary theoretical and methodological lenses. Overall, it is significant to explore the emotional challenges and coping mechanisms of foreigners, especially those who pursue their academic careers during this persistent public health crisis. Therefore, we aim for a more in-depth understanding of mental health of IFMs during the COVID-19 pandemic and provide implications for clinical and counseling practices in a specific geographical and regional context in China.

## 2. Review of the Literature and Theoretical Framework

### 2.1. COVID-19 Pandemic and Psychological Trauma among Socially Vulnerable Populations

The impact of the COVID-19 pandemic on human ecological systems has prevented individuals from pursuing well-being in their lives and has also incited trauma responses [12,13,14]. Psychological traumatic experiences are organized by time: before, during, and after [15]. They start with denial, non-acceptance, and mistrust, followed by acceptance, confusion, and the ramifications of the safety measures arising due to the external environment, and hopefully, they end with transformation and new development after the trauma. More specifically, potential traumatic symptoms and risk factors can occur due to causes, effects, and objects during a pandemic [13,16].

Scholars in public health, medicine, and social and behavioral science have shed light on mental health among humanity, especially psychological trauma among individuals belonging to various social categories, such as class, gender, race, ethnicity, and national origin [12,13,14]. Based on the notions of human geography and comparative and international education (CIE), researchers have investigated the effects of the pandemic on global migration and global student mobility (GSM) and highlighted emotional unrest in the nexus between cultural politics and the intersectionality of these social categories, that is, the power differentials that cause social exclusion and discrimination between locals and foreigners, including educational sojourners, teachers, students, and their families [17,18,19].

Overall, socially vulnerable individuals can experience more severe psychologically traumatic symptoms, especially in lockdowns or quarantines provoked by the COVID-19 pandemic [4,20,21,22,23]. Therefore, the emotional challenges and psychologically traumatic symptoms of individuals in lockdowns can potentially entail psychological and physiological reactions. These symptoms can potentially lead to post-traumatic stress disorder (PTSD), which accompanies long-term negative memories of certain traumatic events [21,24].

### 2.2. Trauma-Informed Care

Scholars have paid close attention to diverse coping mechanisms during the COVID-19 pandemic. From among many conceptual maps, this study considers trauma-informed care (TIC) as its primary theoretical framework to investigate mental health of IFMs in China, as potential psychologically traumatic incidents and risk factors can begin as specific symptoms arising from socially vulnerable individuals’ experiences. TIC is a strength-based framework for clinical practice that has been used to improve well-being. It focuses on responsiveness to the impact of trauma and emphasizes physical, psychological, and emotional safety; connection; and emotional regulation that creates opportunities for survivors to empower themselves [25].

TIC creates a basis for understanding the negative impact of psychological trauma and can help prevent re-traumatization [25,26]. TIC’s three pillars, which create a context for healing, are (a) safety, (b) connections, and (c) managing emotions [26]. Safety relies on consistency, reliability, predictability, and transparency to promote safe environments. However, it also depends on the second pillar (connections), something created to maintain positive interactions. These positive interactions, in turn, help with the third pillar (managing emotions), which improves the ability to control one’s impulses and regulate oneself. These three pillars work together to create a context for healing, but they both impact and are impacted by surrounding human ecological systems [26].

TIC suggests clinical and counseling practices through which clients can view their active roles in diverse socio-cultural and political-economic climates. Counselors can provide appropriate guidelines, perhaps about safety expectations and the availability of legal, financial, and social benefits [25,27,28]. From these perspectives, we anticipate that it would be worthwhile for counselors to encourage IFMs to reflect on cross-cultural values and intercultural communication with locals in certain social, cultural, and political environments in China, thereby pursuing their own social and emotional well-being through acceptance rather than denial. Self-care is always significant in counseling, and mental health counselors should engage in self-reflection and develop counseling and coping skills for ethical practice [29].

### 2.3. Rationale for the Current Study and Research Questions

Over the past two years, a robust body of literature has examined the impact of the COVID-19 pandemic on the human ecological system crisis. Higher education is a prominent sector where scholars provide attention to the emotional and social well-being of diverse stakeholders. It also represents a sizeable professional industry for foreigners staying in China [17]. While numerous nations have normalized the level of the public health crisis and increased their cross-border traffic, China has been fortifying the dynamic Zero-COVID policy. Given this, IFMs in Shanghai, Hangzhou, and Nanjing–located in East China along the Yangtze River Delta region–were under the strict national public health policy and endured persistent short-term and long-term lockdowns, as well as PCR tests, during the Delta and Omicron outbreaks (July 2021–June 2022).

Indeed, Shanghai, Hangzhou, and Nanjing are prominent cosmopolitan and key financial hub cities in China and worldwide. The cities’ precision prevention models were highly recognized, and especially Shanghai could control explosive chains of lethal transmission successfully before the Delta and Omicron outbreaks. Their residents demonstrated “strong collective behaviors by wearing masks regardless of age, gender, occupation, or location in looking out for their neighbor’s wellbeing” [5], ([8], p. 1). Yet, over 28.5 million residents in Shanghai, including foreigners, were in a panic and had to deal with sudden emotional changes such as anxiety, uncertainty, stress, and frustration, because they never expected the nation’s most economically advanced city could be sealed [8].

The lockdown experiences of the IFMs during this specific timeframe can greatly be different from other previous lockdown cases, consisting of the long-term citywide lockdown in Wuhan for 76 days (23 January–8 April 2020), because they were unexpectedly isolated into their own residences or compounds without specific notices and plans to be unsealed. Notably, the case of the long-term Shanghai lockdown (2 March–1 June), which has still been ongoing partially (as of August 2022), demonstrates an illuminating example in which residents were not allowed to order food and gain survival necessities [8]. Thus, they could face an enormous number of emotional challenges and encounter potential traumatic events and risk factors.

Additionally, in general, IFMs holding the global talent visa status could be in honorable positions, contributing to promoting a knowledge-based economy and human capital for China. The Chinese academic society values their scientific knowledge, cultural awareness, and linguistic diversity, expecting the IFMs to raise their higher education institutions’ reputations on a global scale [7,8,30]. Nonetheless, in the current lockdown period, they can be viewed as vulnerable populations because of the lack of emotional and social support from their families and friends, language barriers, and cultural distance [31]. These elements can potentially situate them to become less confident socially, culturally, and psychologically, especially when dealing with absorbing the official documentation about frequent policy changes and utilizing digital technology, which requires them to use Mandarin Chinese [8]. Overall, migrants and educational sojourners in foreign countries already find cross-cultural adaptation challenges amid pandemic lockdowns (e.g., SARS and COVID-19), potentially leading to mental health crises if they lack social and emotional support from their families and acquaintances [4,32,33,34].

At this point, it is significant to recognize that scholars have paid scant attention to the lockdown experiences of IFMs in Shanghai, Hangzhou, and Nanjing during the Delta and Omicron outbreaks and overlooked investigating potential traumatic incidents and risk factors that may hamper them from pursuing social and emotional well-being. It is crucial to explore challenges and opportunities to establish possible coping mechanisms for their mental health and contribute to promoting new knowledge in the field of environmental research and public health. Therefore, based upon the concepts of TIC and the rationale for the study, we ask these primary research questions:

RQ1: What were the experiences of IFMs amid the series of lockdowns during the Delta and Omicron variants?

RQ2: What were the most severe psychological traumatic incidents and risk factors of these experiences?

## 3. Methodology

### 3.1. A Transcendental Phenomenological Approach

We adopted a transcendental phenomenological approach as its primary methodological lens because its purpose is to examine the mental health of IFMs who experienced a series of lockdowns in China during the COVID-19 pandemic [35,36,37]. In general, phenomenology is a choice of methodology rather than an attempt to understand individual experiences of the same phenomenon. In this context, it is useful to grasp the essence of the subjects and objects of the study to investigate ‘what’ and ‘how’ participants experienced emotional unrest [35].

The phenomenological approach is appropriate when exploring socio-cultural and socio-psychological factors affecting emotional unrest or life challenges, including trauma, insomnia, anger, and sorrow, that individuals wish to avoid [36,37]. Notably, transcendental phenomenology entails a concept of “*epoche*” (or bracketing), in which investigators set aside their experiences, as much as possible, to take a fresh perspective” ([35], p. 80). This phenomenological approach investigates psychological aspects of individuals’ experiences of newly forming social phenomena and their emotional states to demonstrate that “everything is perceived freshly, as if for the first time” ([37], p. 34) [38]. From these viewpoints, we believe that scholarship can assist people with challenging feelings about the socio-ecological phenomenon of COVID-19. Thus, we aim to deliver participants’ “true” stories: what they have experienced “in terms of the phenomenon” and how situations influenced their “experiences of the phenomenon” ([35], p. 81).

Additionally, given the qualitative nature of critical social research, we incorporated a community autoethnographic approach into transcendental phenomenology in which authors are members of the participant group within their own study that explains “how a community manifests particular social/cultural issues” ([39], p. 18). Hence, this approach aims to “facilitate ‘community-building’ research practices”, making “opportunities for ‘cultural and social intervention’ possible” ([39], p. 18). To implement a community autoethnographic study, two or more authors come to their research committed to collaborative action. They consider personal reflections, conduct interactive interviews with one another, or combine both methodological approaches [39,40]. Therefore, we explored the lockdown experiences of IFMs during the Delta and Omicron outbreaks in China.

### 3.2. Methodological Limitations

We acknowledge that large samples through a quantitative analysis are beneficial for drawing general perceptions and conclusions about particular populations, providing impartial, accurate, and reliable information. However, this approach is limited to deducing conclusive results about the details of social phenomena or problems because it is constrained by fixed questions and specific relationships between variables, which are usually answered using questionaries [41]. Although adopting a quantitative approach and measuring scales and scores to infer the shared experiences of IFMs, can help understand the traumatic experiences of this specific population, there were limitations to collecting large samples in this rapid socio-ecological system crisis.

For example, before the Delta and Omicron lockdown periods in the subject cities such as Shanghai, Hangzhou, and Nanjing (July 2021–June 2022), many IFMs (approximately 80) from our home institutions actively communicated and shared diverse information via WeChat. However, during the lockdown periods, they barely interacted with one another and hesitated or refused to participate in the current study. Thus, these issues constrained increasing large samples. Meanwhile, qualitative research can inform individuals’ vivid experiences, behaviors, perceptions, and opinions more broadly through in-depth interviews and rich and thick descriptions. In these ways, scholars play influential roles in achieving a better understanding of particular social phenomena [35,42,43]. Therefore, grounded in a qualitative approach, we present a transcendental phenomenological study.

### 3.3. Researchers’ Subjectivities

Since this study involves mental health issues and emotional challenges facing ourselves and other individuals amid a series of lockdowns provoked by the COVID-19 pandemic, it is crucial to discuss researchers’ subjectivities, such as positionality, reflexivity, social representations, and cultural practices, as well as personal emotional states, professional experiences, and qualifications. This will increase trustworthiness and minimize potential biases toward the subjects and objects of the study [39,44]. We, the two authors of this research, are American IFMs in China. We are currently affiliated with institutions of higher education in Shanghai and Hangzhou, which are cosmopolitan and metropolitan cities located in the Yangtze River Delta region.

The first author is an Asian-American man. He holds a Ph.D. in higher education administration, with specializations in CIE and cultural studies in education. The second author is a White-American man. He holds a Ph.D. in applied psychology, with specializations in cross-cultural psychology and intercultural communication. In institutional, organizational, and clinical practices, the first author is certified in physical education, recreation, and dance (PERD); exercise, leisure, and sport (ELS); and global leadership (GL). The second author is certified in teaching English to speakers of other languages (TESOL) and has extensively dealt with cross-cultural training, stress management, and workplace coaching. Accordingly, we have provided services to socially vulnerable populations in both the U.S. and China, especially for temporary residents, migrants, international students, and their families.

We represent ourselves as insiders in this research because of our membership in the inner cultural group of IFMs in China. We, too, have experienced a series of lockdowns in Shanghai and Hangzhou. However, we acknowledge that there are always different interpretations and potential biases toward specific study subjects and objects based on divergent ideologies, social representations, cultural practices, and national origins. Therefore, we admit that we shall mutually interpret the findings by respecting all ethical considerations in terms of trustworthiness [35,42].

### 3.4. Participants

Phenomenologists typically consider a relatively small number of samples but select samples that can provide rich and thick descriptions through which they “interview from 5 to 25 individuals who have all experienced the phenomenon” ([35], p. 81). Accordingly, by adding ourselves to the participant group, we used purposeful and snowball sampling methods to recruit 16 participants during the Delta and Omicron lockdown periods in Shanghai, Hangzhou, and Nanjing. Thus, the current study includes a total of 18 participants [39,40,42]. Simply put, seven participants were from our own institutions, and they introduced us to the remaining participants, who are their close friends. In terms of age, participants range from their 30s to their 50s. There are 13 men and five women. Broadly, their academic fields are science, technology, engineering, and mathematics (STEM); information and communication technology (ICT); business and economics; the social sciences; and the arts and humanities. The length of their academic careers in China ranges from 3 to 12 years (see Table 1).

We conducted semi-structured interviews with participants from Nanjing and Hangzhou during December 2021, initially aimed at investigating the experiences of the Delta outbreak (July–November) in China. Yet, due to the emergence of the Omicron outbreak at the end of 2021 through the middle of 2022, which became severe in Shanghai from February to June, we included participants from Shanghai in the middle of February through early June. Therefore, the interviews were scheduled between 7 December 2021 –5 June 2022. Considering the evolution of the pandemic, the primary individual interviews were conducted via multipurpose social networking platforms and apps (i.e., WeChat and Tencent). Additional formal and informal conversations continued via WeChat. In the meantime, we also conducted interactive interviews, developed reflexive journals, and communicated online regularly. Given this, we captured over 300 WeChat texts that indicate our personal communications with each other and other foreigners (see Table 2).

### 3.5. Data Analysis

We considered a thematic analysis technique [42]. In the initial phase, we openly coded the interview or textual data from WeChat to develop the primary themes. Initially, we contemplated a role of TIC that can promote counseling and clinical practices through the lens of the participants’ lived experiences, namely suggesting guidelines on how to improve clients’ engagement, treatment, adherence, health outcomes, and mental wellness [25]. As aforementioned, TIC entails several core principles, consisting of safety, choice, collaboration, trustworthiness, and empowerment. Hence, we focused on identifying the similarities and differences between our personal reflections (assumptions) and the participants’ lived experiences and perceptions about their lockdown experiences. As members of the inner cultural group, we reviewed the overall textual data, which explicitly demonstrates the interaction of the disease with life changes in real-time, with a particular focus on emotions during a series of lockdowns and PCR tests.

In the next phase, we focused on ensuring the physical and emotional safety of participants and seeking factors that can increase our understanding of TIC and how the participants can grow and comprehend their experiences through their general social and cultural lives beyond their academic lives (e.g., teaching and research) amid long-term lockdowns [26]. Accordingly, we focused on identifying common or differing elements in real-time. We found numerous emergent themes alongside the participants’ own human ecological circumstances and cultural norms, including emotional, political, ideological, occupational, social, economic, cultural, and educational factors [8]. Finally, we reviewed the data multiple times, focusing solely on the specific emotional challenges arising from diverse psychologically traumatic incidents and risk factors, to seek coping mechanisms in the given nature of the phenomenological approach [4,7,10].

Overall, we mutually agreed to locate the participants’ collective ideas and behaviors in their own geographic and regional contexts (i.e., Shanghai, Hangzhou, and Nanjing). As a result, political and ideological factors collapsed due to being less influential: participants barely discussed or acknowledged the government’s political decisions to implement lockdown or social distancing policies. The overall analysis relied on the most common and challenging issues involving IFMs amid long-term lockdowns.

### 3.6. Trustworthiness

Concerning trustworthiness, we considered several vital elements, including ethical approval, audit-trials, member-checking, cross-checking, participant collaboration, and thick and rich descriptions. Initially, it is essential to respect ethical approval in a qualitative inquiry, especially a phenomenological approach that deals with vulnerable individuals’ emotional challenges. Accordingly, we obtained IRB approval from the Research Administration Office at the first author’s home institution. Upon the participants’ consent, we conducted semi-structured interviews and follow-up communications. Protecting participants’ personal identities is crucial. Thus, we used pseudonyms and removed their specific institutional information. By the same token, we also hid our individual identities within the participant group [42].

Furthermore, we conducted an audit-trial by inviting two American scholars and two Chinese scholars in counseling psychology, social psychology, global migration studies, and comparative and international education. They observed the research process and provided feedback regarding data analysis. None of these scholars were involved in this study as participants. Additionally, we conducted member-checking with five participants to validate data and ensure their reliability. We also conducted cross-checking and mutually discussed determining the conclusive findings. We shared the most significant factors that could influence our participants’ emotional challenges and, in the meantime, reflected on our empathies [35].

In phenomenology, investigators could also experience similar phenomena that participants experience. Thus, as an inner cultural group in this study, we considered participants’ collaboration by mutually sharing general lockdown experiences with our participants. We jointly paired with some participants and had continual communication. For example, the first author interacted with one American and one British participant, and the second author interacted with two other American participants [43]. Given the nature of the qualitative approach, it is crucial to offer vivid narratives of individuals’ life stories. Therefore, we considered thick and rich descriptions to promote scholarly conversations about the mental health issues of IFMs during the Delta and Omicron outbreaks in Shanghai, Hangzhou, and Nanjing, China [35].

## 4. Findings

### 4.1. Neuroses about the Omen of Lockdowns

Our participants mentioned the term “omen”, which referred to their intuitions, sensitivities, nervousness, and strong feelings about potential unexpected traumatic incidents, such as a series of lockdowns, in response to hearing about cases in surrounding communities or even neighboring cities. Initially, American-IFM-2, who is staying in Hangzhou, recalled:

“At the end of the spring semester of 2021, I had already read an article on WeChat about the growing cases of the Delta variant in other countries, but I did not think it could happen in China.” Similarly, American-IFM-3, who is staying in Nanjing, told us, “It was near the end of the spring semester of 2020 and continued through the entire summer break. I was in my apartment in Nanjing. My Weibo [social media platform] began alerting me that there were again some feverish patients in this city. However, most locals seemed normal.”

IFMs soon realized that the Delta and Omicron variants could be even more contagious than their predecessors, which greatly influenced their general life experiences in China. For instance, German-IFM-1 testified: “I felt that my own residential areas could be sealed if even one confirmed case was found in surrounding communities”. American-IFM-1 recalled one recent event: “All faculty and students were in the classroom. Suddenly, the school administration announced that we were not allowed to leave. We were isolated on campus for more than 10 h, and key leaders and administrators [Chinese faculty] like deans and their staff members stayed on campus for more than three days.”

When it comes to feelings about lockdowns, American-IFM-6, who stays in Shanghai, recalled an unexpected issue in his apartment building and said, “When my compound was initially sealed, all gates were locked, including emergency doors. I also found that all elevators stopped working. Our building has 25 floors. All of the residents felt so anxious about safety. What if it’s on fire? Everybody was stuck in the building.” Furthermore, Korean-IFM-1 said, “every Korean male must complete mandatory military service for a few years. But at least soldiers can breathe fresh air and go outside; this is a different feeling. I felt very heavy and obsessed with opening the windows and looking outside. All I could do was prepare for courses online and check the time of day again and again.”

Additionally, Turkish-IFM told us, “I was getting crankier and constantly exchanging texts with my colleagues or school administrators about when I would be set free. Now, I am often under pressure because of unforeseen lockdown experiences. The very first thing I do after I wake up is to go downstairs and check if my community is sealed or not.” Other participants such as Japanese-IFM, German-IFM-1, and German-IFM-2 also had similar feelings. They meant that each time when they were isolated in their own places, they felt extreme stress, anxiety, frustration, and obsession with uncertainty amid a series of lockdowns.

Recent studies about lockdowns among expats, including international faculty and student bodies, showed specific risk factors that individuals cannot avoid. Educational sojourners staying in foreign countries could express extreme stress and frustration, which contributes to developing long-term trauma. The particular symptoms entail anxiety, exhaustion, insomnia, and nightmares [4,10,11]. Neurosis as a form of traumatic risk factor is closely intertwined with negative memories. From a human ecological perspective, counselors explore the relationship between individuals’ negative memories and external systemic barriers in which potential traumatic incidences and risk factors coexist in a wide range of political, economic, social, and cultural contexts [45,46]. Therefore, counselors should gain professional knowledge about diverse environmental factors, consult their clients about developing self-coping skills such as humor and mindfulness, and help them look for the bright side in their experiences of sudden ecological and environmental changes.

### 4.2. Exhaustion and Frustration about Persistent PCR Tests

Aside from the omen of the lockdown, one of the most challenging issues arising from a series of lockdowns was the persistent requirement for PCR tests. For example, British-IFM-2 recalled the situation surrounding the Delta variant in Hangzhou at the beginning of the fall semester of 2021: “We were isolated on campus for more than 24 h and probably up to 30 h [depending on the campus locations in Hangzhou] until all individuals on campus had taken PCR tests. We had to sleep on the couches in our own offices. I returned home at night and felt very exhausted.” From late February to early March 2022, the Omicron variant became severe in Shanghai. American-IFM-1 testified:

“On 10 March I returned home and woke up in the morning because all faculty and students were required to have PCR tests. The school initially required us to complete six tests within 15 days. But after I had my third test, my apartment was suddenly in lockdown. The residents in our apartment community were informed that we were initially required to stay home for 48 h and would have two more tests. When everyone tested negative, we would be able to go outside. But it’s frustrating because they sealed the entire community without any notice. We didn’t have food and water or other necessities. And frankly speaking, it wasn’t for 48 h. We were isolated in the building for more than 60 h. For the past few weeks, I’ve had tests more than 10 times.”

British-IFM-1, who lives in Shanghai, also shared her story and feelings about the constant PCR tests: “That’s expensive. You sometimes need to take transportation to go somewhere, right? I’m okay with the subway, but for me, it’s like I have to pay for the test tube.” Given this, she continued to express her anxiety: “The subway line also goes to the hospital, where you encounter a lot of sick people. You might catch something else… I’ve subjected myself to the possibility of catching the flu, a cold, or something else.”

Meanwhile, the middle of March saw a severe outbreak of the Omicron variant in other metropolitan cities. For example, British-IFM-3 in Hangzhou said, “It’s uncomfortable and it’s expensive, you know, because if you want a free test, you must come to the campus. If the campus is too far away or if you don’t want to go to the campus because your district is not medium risk, you’re doing it elsewhere several times and it’s 40 RMB [6 USD] each time.” Korean-IFM-1, Korean-IFM-2, and Japanese-IFM also commonly pointed out that traveling to the campus is not only a serious problem but also staying in lines for a long time to have PCR tests makes them feel extremely uncomfortable and anxious due to the growing confirmed cases in their cities.

Core TIC principles entail safety, choice, collaboration, trustworthiness, and empowerment to support vulnerable populations who cannot sustain themselves financially, socially, and culturally [25]. To link some of these principles in the Chinese national context, it is significant to understand the environmental, geographical, and cultural elements of East China, namely the Yangtze River Delta region, and their societal values, such as collectivism. Briefly, residents in the area have long practiced stronger collectivism inherited by the historical rice-farming community than residents in North or Northwest China (e.g., Beijing and Xian) inherited by the wheat-farming community [47,48].

Shanghai, Hangzhou, and Nanjing belong to the historical rice-farming society according to the environmental, geographical, and cultural factors [49]. Residents in the Yangtze River Delta region could demonstrate a level of sacrifice to promote their neighbors’ wellness (e.g., mask use, social distancing, and community volunteerism) regardless of age, gender, occupation, and socio-economic status [5]. A recent study also found that rice-farming societies worldwide and in China had fewer coronavirus deaths than other cultural regions [50]. They also tend to conform to the strict national public health policy, establishing safety and emergency networks during a series of lockdowns and PCR tests, even though they face diverse forms of emotional challenges [5,8].

Promoting collective resilience with host country members is significant. Thus, counselors should consider ways to establish safety networks with locals and develop cultural coaching programs. In these ways, foreigners could gain more fruitful social and emotional support and reduce emotional challenges. As aforementioned, IFMs holding global talent visa status in the nation’s most economically advanced cities in the Yangtze River Delta region could receive honor and privilege as the Chinese academic society highly values their scholarship. Yet, they could be vulnerable without fruitful social support during the lockdowns. However, it could be an opportunity to reduce language barriers and cultural distance if they respect the local customs, rules, and laws and communicate with their host country members (e.g., domestic faculty, students, and compound committee members) [8,51]. In so doing, domestic and foreign parties can co-construct safety networks, mutual rapport, and trust, creating pathways to empowerment for various vulnerable populations.

### 4.3. Leisure Constraints: Experiencing Burnout and Witnessing Dropout

One of the most common emotional challenges faced by IFMs, as identified in the current research, was related to leisure constraints. Among diverse psychologically traumatic risk factors, burnout was a critical issue that IMFs could not deal with while pursuing emotional and social well-being. For these reasons, some witnessed dropout, with colleagues terminating their contracts in the middle of a semester and leaving for their home countries or for institutions in safer areas. For example, American-IFM-2 testified:

“People value diverse leisure activities, including card and board games, theater, traveling, drinking, and eating at restaurants. It is important for people to get together, build rapport, network, and enjoy leisure activities… I had a close colleague who recently left my school after the Delta outbreak because he couldn’t take part in outdoor activities, and he is in Beijing now… Because Beijing is the capital city and symbolically important for China, he thought it would be safer and that the government wouldn’t seal the entire city. But you know that there is no safe place in this time of public health crisis.”

Additionally, British-IFM-1 emphasized that spending time with colleagues and friends via leisure activities is extremely important. However, during lockdowns, under the Chinese government’s social control, it was challenging to get together with others. She said, “I usually hung out with my colleagues or friends after work. This is very important in our international community. Recently, my close colleague, who worked in another school, terminated her contract and left for her home country because of the limited opportunities to enjoy her social and cultural life outside.” Additionally, American-6 recalled, “local communities created diverse social events, activities, and festivals for foreigners, who also enjoyed the hospitality of these communities through such programs as dance, weightlifting, ping-pong, and other sports. However, all leisure activities and outdoor programs were stopped.”

Social and health psychologists addressed that leisure constraints frequently occur during pandemic periods (i.e., SARS and COVID-19) [8,13,14,52]. Human ecological systems, such as macrosystems intersecting with cultural environments and individuals’ lifestyles (i.e., leisure and physical activities), are significantly malfunctioning. The drastic changes pertain to historical trauma, past epidemiological quarantine experiences, and gainful experiences from these recent events [45,52].

From these perspectives, the participants’ voices can account for the human ecological system crisis. However, we could see potential opportunities for counselors to provide comprehensive advice on how to find alternatives for reducing leisure constraints and increasing professional role identity. In practice, IFMs are teachers and advisors of their domestic students. They can concentrate on engaging in various professional, social, and cultural activities via online platforms, sharing their talents to support their schools and surrounding communities. They can also engage in public and social services with their domestic colleagues, promoting collective resilience such as mutual social and emotional support for their students and community members and prosocial behaviors among themselves [51,53].

Perhaps they could provide online mentoring and tutoring services, social media/blogging activities, and cultural exchange programs based on their expertise and knowledge. These alternatives could include foreign language tutoring or exchanges, art activities, literature, films, and book reviews, among others. In so doing, they could promote their professional role identities, feel a sense of belonging, release stress about their leisure constraints, and develop new social and cultural lifestyles. As TIC is a strength-based approach, counselors and clinical practitioners can help their clients sustain themselves at a time when they are encountering physical and emotional safety concerns. Dynamic regulations and alternatives can develop individuals’ opportunities to negotiate for survival. These can create pathways to empowerment and promote environmental sustainability [25].

### 4.4. Concerns about Families and Friends: Separation, Illness, Loss, and Grief

As members of the inner cultural group–that is, as colleagues who have been closely interacting and communicating over the past two years–both the participants and we, the investigators of this study, witnessed numerous emerging challenges and difficulties posed by the Chinese national public health policy, which has caused separation among families and friends for years. We have directly heard negative news, for example, about illnesses in the IFM community. Some have witnessed illnesses among their international colleagues’ family members in their native countries. In the most extreme cases, some participants directly experienced or witnessed their colleagues’ losses and grief while enduring a series of lockdowns.

A Canadian-IFM who is staying in Nanjing shared a colleague’s painful memory: “On 15 March my colleague heard his father had been hospitalized due to a diagnosis of stomach cancer and diabetes. All these issues impacted his mentality. I heard he passed out, alone for several hours. This affected my emotions too. I would’ve come over so we could cry together, but I couldn’t because of the lockdowns.” Similarly, American-IFM-2 testified, “While I’ve been in China, one of my family members became infected [with the Coronavirus]. It was a very hard time. I felt depressed and stayed up through the night to deal with my emotions.” Meanwhile, British-IFM-1 discussed homesickness: “I often feel very angry. You have to talk with your family, especially when you’re sick or your family’s sick. It’s very important for me to talk about my feelings with my family in person. It’s been over three years since I’ve seen my parents and sister, or my friends, even if they’ve been unwell. I miss home, but at this time, I cannot do anything.”

Additionally, some of our participants had recently experienced mental crises due to their own family well-being. For example, American-IFM-2, American-IFM-4, American-IFM-5, British-IFM-2, and British-IFM-3 are colleagues in Hangzhou. As a selective inner cultural group [Anglophone culture], they could naturally become socialized. They recently lived through extreme situations in which their universities had just a single COVID case but required all faculty and students to return to campus within 3 to 4 h to get PCR tests, which they could only do on campus. They were parents who wanted to take care of their children but were instead forced to isolate on campus. Representing the group, American-IFM-2 said:

“We’re from Western countries and Thanksgiving is a special family occasion. We had to get PCR tests on Thanksgiving night at 11:30 p.m. A potential lockdown did not allow us to go outside. We were isolated on that Thanksgiving night. The emotional stress of lockdown forced all IFMs to prepare by bringing extra clothes and supplies to leave at the university in case of a sudden lockdown. We were forced to stay on campus for days. When the outbreak happened, our entire families, including toddlers, were forced to stay at home in small apartments and couldn’t take part in any outdoor activities. Even though they were under the weather, they couldn’t go to hospitals.”

American-IFM-1 and American-IFM-3 have been married for five years and are currently living in different cities. They had been about to have a baby but recently lost it. American-IFM-1 and American-IFM-3 shared this aspect of their family story:

American-IFM-1: “Back in October 2021, I heard I was going to be a father. Like other parents, it was the happiest moment of my life, and of my wife’s. My wife and I prepared a new home and bought gifts and jewelry for the baby. But just two days after I returned to Shanghai from Nanjing, my wife contacted me and said the fetus was in critical condition, which could impact her life. We lost the fetus around 10 a.m. on March 19. For a month afterwards, I had extreme feelings of stress, exhaustion, insomnia, nightmares, and negative memories, all while enduring a series of lockdowns.”

American-IFM-3: “Before the COVID-19 outbreak, we could be together physically. Although we experienced a series of short-term and long-term lockdowns and persistent PCR tests, we could deal with any challenges. But then this happened. This is the most painful memory of our lives.”

WeChat message from American-IFM-1′s mother: “I don’t know how to comfort my children. Because Omicron has spread so severely, it’s impossible to visit China. I must confess that I had no choice but to watch you with my heart and feelings because I couldn’t share this sorrow with you. Both of you in China are suffering heartache. And right now, I just hope that both of you can achieve some stability and return to a healthy daily life soon. My daughter in-law’s lips were swollen, so I couldn’t stop crying. For parents who have raised children, the pain of their children is their own pain, and they want to accept that pain if they can. I understand that everyone has suffered a lot physically and mentally this time. I think of it as having a bad dream while traveling together. I hope that your health does not deteriorate.”

Finally, American-IFM-7 shared his general perception of the impact of the COVID-19 pandemic on lifestyle changes and described his specific emotional status and empathy regarding the holistic well-being of IFMs’ families and friends:

“As IFMs with family overseas, we have a clear understanding of the COVID-19 pandemic experiences thanks to family members in China and in our native countries, where the pandemic has been more severe though it now seems to be slowly returning to “normal.” This means that we are all aware of alternative experiences and COVID-19 policies or softer restrictions and often yearn to go back home to see family and friends. We also know many people who have tested positive for COVID-19 and overcame it. This makes many IFMs feel a loss of hope and, in some instances, regret: a sense that the grass is greener on the other side.”

Recent evidence from COVID-19 research showed growing concerns related to PTSD impacted by families and friends. Notably, individuals’ loss and grief are unavoidable and persistently happening during the public health crisis [1,4]. Our current study also revealed the loss and grief experiences. Counselors should use TIC to increase a more in-depth understanding of their clients’ negative memories and psychological trauma provoked by the long-term lockdowns. They can work to reduce re-traumatization through three pillars such as safety, connections, and emotional management [26]. They can also suggest guidelines about safety and harmony that influence their clients on how to rethink and reframe their positive emotions. Counselors should encourage their clients to establish continual communication networks with their families and friends; they can demonstrate active challenges to overcome diverse emotional barriers.

## 5. Discussion and Implications

IFMs in the current study encountered various psychologically traumatic incidents and risk factors amid a series of lockdowns during the Delta and Omicron outbreak periods. In clinical and counseling practice, trauma-informed care (TIC) is crucial for treating individual clients during COVID-19 [54]. A TIC approach is essential to ensure clients are supported before, during, and hopefully after the COVID-19 crisis. TIC is about creating a culture built on safety, connection, and emotional management, which can aid in creating TIC structures at the start of systems and can be integrated with multiple human ecological systems. The goal is to prevent more intense trauma symptoms and re-traumatization. Counselors offering the TIC approach to all clients do so with the understanding that most people have experienced some type of trauma, including loss and bereavement. As the pandemic impacts IFMs in China, it is essential to embrace TIC as it outlines vicarious trauma and triggers responses for both counselor and client [25].

As aforementioned, self-care is always vital in counseling. For ethical practice, counselors should engage in self-reflection, counseling, and coping skills [25]. TIC also emphasizes physical and emotional safety and encourages precautions to prevent re-traumatization [25]. It is crucial to practice all safety guidelines while providing counseling, especially with increased telehealth services. Counselors are encouraged to gain continuing education on best practices and appropriate implementation of telehealth services. Hence, TIC also promotes opportunities for clients to rebuild control by creating safe environments [28].

For foreigners such as IFMs or international students staying in China, counselors can create spaces for consistency, reliability, honesty, and transparency, all of which can be challenging when the surrounding environment provides the opposite qualities. However, counselors’ specific skills in building therapeutic relationships will contribute to forging connections and building support systems [25]. Counselors can be reliable within unreliable spaces. With a safe space and strong relationship, counselors can help clients unpack the overwhelming emotions as they face an unknown future and engage with challenging feelings they may initially want to avoid. Emotional avoidance can hinder well-being and prevent individuals from overcoming psychological trauma [27].

For IFMs, one of the challenging issues was leisure constraints which entailed burnout and potential dropout factors. In this scene, it can be helpful to explore new leisure activities and develop new ways of engaging with the support systems previously gained from activities that became inaccessible during isolation. Leisure activities play a critical role in well-being during COVID-19, and that outside activity before isolation supported coping skills during quarantine [55]. Participants demonstrated that the loss of leisure activities impacted their mood. However, if they had been active before, participants were more likely to engage in new activities that promoted meaning. Counselors can consider this when helping clients develop new coping and leisure activities and connect through multiple systems.

For those who lost loved ones, such as families and friends, specific interventions could include mindfulness techniques that possibly integrate with TIC. Specifically, counselors can use grounding techniques to create safety and decrease the client’s use of avoidance strategies to cope with unwanted thoughts and emotions related to trauma. Counselors can work with clients to increase the acceptance of, or willingness to experience, previously uncomfortable thoughts. They can move through denial by establishing safety-building skills [56]. It is instrumental to merge grounding interventions or mindfulness with a TIC conceptualization to help counselors and clients understand how they both influence and are influenced by their environments and how they can promote well-being.

Counselors are trained to serve as leaders, advocating and collaborating to promote systemic change [57]. Many institutions and governing bodies need to address these areas as well. Still, counselors can work as advocates to aid in developing TIC policy that focuses on building health support systems and promoting healthy leisure activities. For example, social distancing guidelines can come with suggestions for healthy leisure activities; these could involve social media, family-specific opportunities, counseling, or outdoor activities for larger groups that follow safety guidelines [58]. Counselors can use resilience to help understand clients during COVID-19 and to develop their own advocacy for healthy structures.

Counselors can integrate resilience to explore social and cultural considerations and systemic barriers that impact IFMs’ well-being. For example, individual-level interventions may include mental health or life history assessments in which counselors should consider if their clients previously experienced mental health issues. It can help determine the origins and developmental course of a trauma response. They can also assess how COVID-19 has impacted family, peers, friends, and the local community and help develop support systems [55]. The mesosystemic level of resilience accounts for the quality of relationships within different contexts (e.g., the relationships between family members, work needs, geographical location, interaction with illness, and policy) [55].

Counselors can create helpful support structures by strengthening the relationships between clients (i.e., IFMs) and community members and finding other avenues for connection within policy guidelines. Due to social distancing, this will require creativity and advocacy, which depend on issues of equity. There is also a macro-level influence from environments of which the client is not directly a part (e.g., the government). Counselors can use this to increase the understanding and awareness of discrimination, prejudice, economics, and policy concerning a client’s trauma responses. Specifically, it can help address fears and concerns about certain human ecological security (e.g., public administration, social agencies, cultural organizations, and educational institutions) in relation to family structures and social support systems [59]. Therefore, TIC can help provide a timeframe and normalize feelings associated with COVID-19. Given the historical element of trauma, this may also address past experiences or traumas among IFMs in this research that may have emerged due to current conditions or anniversary reactions.

Additionally, TIC can be more conceptualized through psychological research in understanding how people’s emotions can affect their well-being in consecutive lockdown periods. TIC can also expand in cultural spheres because each national or culture-sharing group represents different ideologies, norms, beliefs, and life practices. Our empirical findings demonstrated that participants who came to China were well-respected as foreign talent scholars in the Chinese national context of Confucianism; scholars have long- developed moral and social systems [60,61]. However, these foreign talent scholars in the nation became vulnerable due to their foreigner status, language barriers, perceived discrimination, and cultural distance [31].

Participants’ concerns about the omen of lockdowns, frustrations about consistent PCR tests, and self-imposed isolation made them difficult to communicate with others, significantly impacting their emotional status, which often systemically prevent them from integration into local cultures under the Chinese national public policy. Given this, TIC can encourage these populations to integrate their cultural practices into the local or host communities, demonstrating acceptance over denial. This implication is crucial because promoting mutual international awareness and understanding can be accomplished when foreigners and host country members co-constructively establish safety networks [4,8]. Overall, TIC is a strength-based framework in social and health psychology aimed at creating pathways to empowerment for vulnerable populations. This study provides implications for establishing potential coping mechanisms for IFMs. Therefore, we suggest clinical and counseling practices for larger audiences and stakeholders in higher education administration and social welfare. We emphasize the importance of TIC to help promote safe environments for educational sojourners and migrants.

## 6. Limitations and Future Research

Some limitations of this research are the constantly changing nature of the pandemic and the time limitations of any related findings. Thus, future research should include a longitudinal study and explore the post-pandemic response. We also did not gather past trauma experiences; therefore, future research could explore how past trauma or ACE (adverse childhood experience) scores impact post-pandemic response. Finally, it would also be beneficial to explore how clients can seek ways to overcome their emotional challenges and actively work to establish their own coping mechanisms. Hence, future research should consider how to build resilience through social and emotional support from locals, as well as prosocial behavior, to develop mental toughness in this time of public health crisis [53]. This study also calls on future scholars to explore the impact of the COVID-19 pandemic on children and the creation of sustainable trauma-informed teaching pedagogy for K-12 and higher education. Finally, this study used a qualitative approach with a relatively small sample size. Therefore, future scholars should consider adopting a quantitative approach to provide more generalizable, impartial, accurate, and reliable information, promote methodological rigor, and expand existing knowledge into global scholarship.

## 7. Conclusions

This transcendental phenomenological study explored psychologically traumatic incidents and risk factors among IFMs who experienced a series of lockdowns during the periods of the Delta and Omicron COVID-19 variant outbreaks. We perceive that it is crucial to observe specific political, economic, and social systems and uncover more positive influences on challenges. For example, how do governments, business enterprises, social agencies, and educational institutions impact trauma responses, and what kinds of social and cultural biases come into play and help normalize national systems at both micro and macro levels? With a more extensive system, it is crucial to consider how policies directly impact an individual’s mental health–specifically, how a vulnerable population is affected by policy changes during COVID-19. To sum up, throughout the TIC process, this study provides considerations for working with clients in the context of COVID-19. It promotes building a culture of support, as every person and every system has been impacted by the pandemic. TIC could help reduce the psychosocial impact of COVID-19. Continued research on experiences can help outline the best practices for counselors before, during, and after a global disaster.

## Figures and Tables

**Table 1 ijerph-19-11057-t001:** Demographic Information of the Participants.

Pseudonym	Age	Gender	Academic Field	Location	Length of Academic Career in China	Self-Reported Chinese
American-IFM-1	40s	Male	Social Science	Shanghai	4 Years	Spoken: Intermediate Reading: Elementary
American-IFM-2	30s	Male	Social Science	Hangzhou	6 Years	Spoken: Advanced Reading: Advanced
American-IFM-3	30s	Female	ICT	Nanjing	4 Years	Spoken: Advanced Reading: Advanced
American-IFM-4	30s	Male	Humanities and Arts	Hangzhou	3 years	Spoken: Intermediate Reading: Elementary
American-IFM-5	30s	Male	STEM	Hangzhou	8 years	Spoken: Elementary Reading: Elementary
American-IFM-6	40s	Male	Social Science	Shanghai	3 years	Spoken: Advanced Reading: Intermediate
American-IFM-7	40s	Male	Social Science	Hangzhou	9 years	Spoken: Advanced Reading: Advanced
British-IFM-1	40s	Female	Business Economics	Shanghai	9 Years	Spoken: Intermediate Reading: Elementary
British-IFM-2	30s	Male	STEM	Hangzhou	7 years	Spoken: Intermediate Reading: Elementary
British-IFM-3	30s	Male	Business Economics	Hangzhou	12 years	Spoken: Advanced Reading: Advanced
British-IFM-4	40s	Male	Social Science	Shanghai	12 years	Spoken: Intermediate Reading: Intermediate
Canadian-IFM	40s	Female	Humanities and Arts	Nanjing	8 Years	Spoken: Advanced Reading: Advanced
German-IFM-1	40s	Male	Business Economics	Shanghai	5 Years	Spoken: Intermediate Reading: Elementary
German-IFM-2	40s	Female	STEM	Hangzhou	10 Years	Spoken: Elementary Reading: Elementary
Japanese-IFM	50s	Male	Humanities and Arts	Nanjing	4 Years	Spoken: Advanced Reading: Advanced
Korean-IFM-1	40s	Male	STEM	Shanghai	4 Years	Spoken: Intermediate Reading: Intermediate
Korean-IFM-2	40s	Female	Humanities and Arts	Shanghai	8 Years	Spoken: Advanced Reading: Advanced
Turkish-IFM	40s	Male	Humanities and Arts	Shanghai	5 Years	Spoken: Elementary Reading: Elementary

**Table 2 ijerph-19-11057-t002:** Interview Protocol for Participants.

**Demographic Information**
(a) What is your nationality? (b) How old are you? (c) What is your academic field? (d) How long have you been staying in China? (e) What are your Chinese literacy and proficiency levels? (f) Where did you stay during the lockdowns?
**General Pandemic Experience and Emotional Challenges**
(a) Please share the overall pandemic experience in China. (b) Please describe your emotion midst a series of lockdowns in China. (c) Please share your stories about the most traumatic incidences and risk factors. (d) Please share your stories about the places where you directly experienced or witnessed.
**Communication and Support**
(a) Please describe how you communicated with your family during the lockdown. (b) How did the university or other local communities help you during the lockdown? (c) How did you communicate with other international faculty members and support one another during the lockdown? (d) Were there any other people who communicated with you and supported you during the lockdown?
**Reflection on Traumatic Experience**
(a) Please describe your emotion during the lockdown broadly again. (b) Did you feel any emotional unrest or discomfort? (c) What have you learned from the pandemic experience?
**Advice and Suggestion**
(a) Describe your fundamental needs while you are currently staying in China. (b) Provide any useful information that the investigators should know about your current life.

## Data Availability

Data is not publicly available due to anonymity concerns. Readers interested in the data can contact the first or the corresponding author upon reasonable request.

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
