# Peer review of "Trauma-Informed Care: A Transcendental Phenomenology of the Experiences of International Faculty during the Delta and Omicron Variant Outbreaks in East China"

_ijerph, 2022, doi:10.3390/ijerph191711057_

Round 1

Reviewer 1 Report

This study investigates foreign scholars' experiences and attitudes towards Covid-19 lockdown issues in several Chinese cities. The topic is important and has enough value to show international readers about this cohort. However, I would like to suggest the authors make some minor changes before accepting the paper. 

1. It is essential to provide more content about ethical approval. The current research design does not offer enough information about ethical issues. For example, where did you get the ethical approval? 

2. What is the role of your theory in the analysis? You mentioned thematic analysis but did not claim how did you use the theory in the analysis. 

3. It is better to link the analysis with your theoretical concepts in the finding part. May add some references to show the connections between data and theory.

4. In the discussion, it is better to say something more about the theoretical implications based on your empirical research.

Good luck with the revision.

Author Response

Trauma-Informed Care: A Transcendental Phenomenology of the Experiences of International Faculty during the Delta and Omicron Outbreaks in East China

Responses to reviewer comments

Authors’ General Response:

We thank the assigned editor and two reviewers for allowing us to take the minor revision work. We also appreciate their sincere comments and scholarly advice, as well as their encouragement to improve this manuscript. Although their decision was a minor revision, we made great efforts to improve the previous version of the manuscript and bolstered both theoretical and methodological rigors. Accordingly, we have carefully considered each reviewer’s comments and have revised the manuscript. Initially, we modified our title slightly as we extensively modified the previous manuscript version. Thus, the new title is intended to provide readers with a simple and clear picture of the nature of the study: “Trauma-Informed Care: A Transcendental Phenomenology of the Experiences of International Faculty during the Delta and Omicron Outbreaks in East China”.

Additionally, we reduced redundancy and replaced some vocabulary, phrases, and ambiguous clauses. We also used active voices as much as possible to deliver meanings. Accordingly, we also checked other minor grammatical and punctuational errors. We also respect reviewers 1 and 2 and their different opinions about the methodological considerations, especially reviewer 2’s comment about a quantitative approach that may be more appropriate for the current study. Given this, we continued considering a qualitative approach. Yet, we significantly bolstered the methodological rigor. We will respond to the reviewer’s comment in detail in the following responses. While we were revising the manuscript, we added much more relevant literature to improve the quality of this study:

Bai, Q., & Nam, B. H. (2020a). Capitalism and reproduction in the new museology: Chinese

cultural heritage conservation and promotion at the Metropolitan Museum of Art. Journal of Arts Management, Law and Society, 50(405), 267-282.

Bai, Q., & Nam, B. H. (2020a). Where ‘West Meets East’: Cross-cultural discourses regarding

the Chinese arts collections at the Metropolitan Museum of Art. Identities. 1-20.
Ellis, C., & Adams, T. E., & Bochner, A. P. (2011). Autoethnography: An Overview. Forum Qualitative Socialforhung /Forum: Qualitative Social Research, 12(1), 1-14.

MacDonald, S., & Headlam, S. M. (2014). Research method handbook. Manchester: CLES.

Peters, M. A., & Besley (2018). China’s double first-class university strategy: 双一流,

Educational Philosophy and Theory, 50(12), 1075-1079.

Talhelm, T. (2020). Emerging evidence of cultural differences linked to rice versus wheat

agriculture. Current Opinion in Psychology, 32, 81-88.

Talhelm, T., & English, A. S. (2020). Historically rice-farming societies have tighter social

norms in China and worldwide. Proceedings of the National Academy of Sciences, 117(22), 19816-19824.

Talhelm, T., Lee, C. S., English, A. S., & Wang, S. (2022). How rice fights pandemics: Nature-

Corp-Human Interactions shaped COVID-19 outcomes. Personality and Social Psychology Bulletin, 01461672221107209.

Zahavi, D. (2019). Phenomenology: The basics. London: Routledge.

We created a few sub-headings in the literature review and theoretical framework and methodology sections. We will describe the overall revision process in detail in the following responses. Overall, every new or modified text and paragraph is colored in red in the manuscript in order to help identify the changes that have been made. We appreciate the reviewers’ insights as they have helped us produce a substantial manuscript.

______________________________________________________________________________

Reviewer #1

This study investigates foreign scholars' experiences and attitudes towards Covid-19 lockdown issues in several Chinese cities. The topic is important and has enough value to show international readers about this cohort. However, I would like to suggest the authors make some minor changes before accepting the paper. 

  1. It is essential to provide more content about ethical approval. The current research design does not offer enough information about ethical issues. For example, where did you get the ethical approval? 
  2. What is the role of your theory in the analysis? You mentioned thematic analysis but did not claim how did you use the theory in the analysis. 
  3. It is better to link the analysis with your theoretical concepts in the finding part. May add some references to show the connections between data and theory.
  4. In the discussion, it is better to say something more about the theoretical implications based on your empirical research.

Good luck with the revision.

Authors’ Response:

We thank Reviewer 1 for his/her scholarly advice. Initially, we acknowledge that it is essential to provide a more specific contextual background of the methodology, especially ethical approval and potential issues that may be arisen. Accordingly, we created a sub-heading, ‘Trustworthiness’, in the last part of the methodology section and described several vital elements to improve our methodological rigor, including ethical approval, audit-trial, member-checking, cross-checking, participant collaboration, and thick and rich descriptions. Notably, we discussed the ethical approval and issues and stated that “we obtained IRB approval from the Research Administration Office at the first author’s home institution.” Accordingly, we also added the Institutional Review Board Statement to the current manuscript that will provide more specific information such as the institution, the IRB protocol number, and the project title.

In terms of the following question about the role of our theory in the analysis, we described key concepts of Trauma-Informed Care (TIC). We linked these concepts to the existing texts in the data analysis section. Concerning the next following question about the findings section. We extensively revised by adding much more relevant literature to apply theoretical concepts to each theme appropriately. Accordingly, we expanded our theoretical implications based on our empirical findings.

______________________________________________________________________________

Reviewer 2 Report

The researchers have made a great effort to be thankful for, but there are some suggestions:

1- The idea of research is usual and I studied a lot and there is no scientific addition

2- The sample is too small

3- The study is qualitative and it was better to apply a scale and convert the results into quantitative scores

4- try to abstract the theoretical part of the study

Author Response

Trauma-Informed Care: A Transcendental Phenomenology of the Experiences of International Faculty during the Delta and Omicron Outbreaks in East China

Responses to reviewer comments

Authors’ General Response:

We thank the assigned editor and two reviewers for allowing us to take the minor revision work. We also appreciate their sincere comments and scholarly advice, as well as their encouragement to improve this manuscript. Although their decision was a minor revision, we made great efforts to improve the previous version of the manuscript and bolstered both theoretical and methodological rigors. Accordingly, we have carefully considered each reviewer’s comments and have revised the manuscript. Initially, we modified our title slightly as we extensively modified the previous manuscript version. Thus, the new title is intended to provide readers with a simple and clear picture of the nature of the study: “Trauma-Informed Care: A Transcendental Phenomenology of the Experiences of International Faculty during the Delta and Omicron Outbreaks in East China”.

Additionally, we reduced redundancy and replaced some vocabulary, phrases, and ambiguous clauses. We also used active voices as much as possible to deliver meanings. Accordingly, we also checked other minor grammatical and punctuational errors. We also respect reviewers 1 and 2 and their different opinions about the methodological considerations, especially reviewer 2’s comment about a quantitative approach that may be more appropriate for the current study. Given this, we continued considering a qualitative approach. Yet, we significantly bolstered the methodological rigor. We will respond to the reviewer’s comment in detail in the following responses. While we were revising the manuscript, we added much more relevant literature to improve the quality of this study:

Bai, Q., & Nam, B. H. (2020a). Capitalism and reproduction in the new museology: Chinese

cultural heritage conservation and promotion at the Metropolitan Museum of Art. Journal of Arts Management, Law and Society, 50(405), 267-282.

Bai, Q., & Nam, B. H. (2020a). Where ‘West Meets East’: Cross-cultural discourses regarding

the Chinese arts collections at the Metropolitan Museum of Art. Identities. 1-20.
Ellis, C., & Adams, T. E., & Bochner, A. P. (2011). Autoethnography: An Overview. Forum Qualitative Socialforhung /Forum: Qualitative Social Research, 12(1), 1-14.

MacDonald, S., & Headlam, S. M. (2014). Research method handbook. Manchester: CLES.

Peters, M. A., & Besley (2018). China’s double first-class university strategy: 双一流,

Educational Philosophy and Theory, 50(12), 1075-1079.

Talhelm, T. (2020). Emerging evidence of cultural differences linked to rice versus wheat

agriculture. Current Opinion in Psychology, 32, 81-88.

Talhelm, T., & English, A. S. (2020). Historically rice-farming societies have tighter social

norms in China and worldwide. Proceedings of the National Academy of Sciences, 117(22), 19816-19824.

Talhelm, T., Lee, C. S., English, A. S., & Wang, S. (2022). How rice fights pandemics: Nature-

Corp-Human Interactions shaped COVID-19 outcomes. Personality and Social Psychology Bulletin, 01461672221107209.

Zahavi, D. (2019). Phenomenology: The basics. London: Routledge.

We created a few sub-headings in the literature review and theoretical framework and methodology sections. We will describe the overall revision process in detail in the following responses. Overall, every new or modified text and paragraph is colored in red in the manuscript in order to help identify the changes that have been made. We appreciate the reviewers’ insights as they have helped us produce a substantial manuscript.

Reviewer #2

The researchers have made a great effort to be thankful for, but there are some suggestions:

1- The idea of research is usual and I studied a lot and there is no scientific addition

2- The sample is too small

3- The study is qualitative and it was better to apply a scale and convert the results into quantitative scores

4- try to abstract the theoretical part of the study

Authors’ Response:

We appreciate Reviewer 2 for his/her critical questions about our chosen subject and research topic, methodological approach, and theoretical interpretation. Concerning the overall questions, we thoroughly considered the reviewer’s viewpoints. However, we decided to continue using a qualitative approach, as Reviewer 1 provided positive feedback about the general methodological considerations and conclusive findings. Yet, we fully contemplated bolstering the methodological rigor and created a few more sub-sections both in the main review of the literature and theoretical framework and methodology sections, consisting of (a) 2.3. The Rationale for the Current Study and Research Questions; (b) 3.2. Methodological Limitations; and 3.6. Trustworthiness.

More specifically, concerning the first question about the idea of research that seems usual and the lack of scientific addition, we respect Reviewer 2’s opinions. Thus, we created “The Rationale for the Current Study and Research Questions” section and fully described the contextual background of the study, stated research problems, gaps, and significance of the investigation thoroughly, and attempted to convince readers why our current study can significantly be different from other lockdown cases.

Notably, regarding the following methodological considerations, we acknowledged that the sample size is small, and a quantitative study can help deduce conclusive results in specific ways. Thus, we created the methodological limitations, discussed both pros and cons of the quantitative approach, and revealed our authentic challenges in increasing the sample size during the specific Delta and Omicron lockdown periods. In the meantime, we also described the positive aspects of a qualitative approach that can inform individuals’ vivid experiences, behaviors, perceptions, and opinions more broadly through in-depth interviews and rich and thick descriptions. In contrast, the quantitative approach is limited to deducing conclusive results about the details of social phenomena or problems because it is constrained by fixed questions and specific relationships between variables, which are usually answered using questionaries.

In terms of the relatively small sample size and other essential qualitative factors that could bolster methodological rigor, in the “Participant” section, we stated that phenomenologists typically consider a relatively small number of samples but select samples that can provide rich and thick descriptions through which they “interview from 5 to 25 individuals who have all experienced the phenomenon” (Creswell, 2013, p. 81). Still, we respected Reviewer 2’s viewpoints about the methodology. Hence, we endeavored to improve the methodological rigor by incorporating a community autoethnographic approach into transcendental phenomenology. Given this, we described in the “A Transcendental Phenomenological Approach” section:

…given the qualitative nature of critical social research, we incorporated a community autoethnographic approach into transcendental phenomenology in which authors are members of the participant group within their own study that explains “how a community manifests particular social/cultural issues” (Hughes & Pennington, 2017, p. 18). Hence, this approach aims to “facilitate ‘community-building’ research practices”, making “opportunities for ‘cultural and social intervention’ possible” (Hughes & Pennington, 2017, p. 18). To implement a community autoethnographic study, two or more authors come to their research committed to collaborative action. They consider personal reflections, conduct interactive interviews with one another, or combine both methodological approaches (Ellis, 2011; Hughes & Pennington, 2017).

Accordingly, we mortified the “Participants” section and indicated that the total sample number is 18. We also specifically described our involvement in the participant group. We further depicted additional validity issues in the “Trustworthiness” section. Finally, concerning the theoretical rigor, as Reviewer 1 also mentioned, we applied the chosen conceptual maps to both the “Findings” and “Discussion and Implication” sections.

Overall, we have included much of the details raised by both reviewers to better position the current study within scholarship on the impact of COVID-19 on higher education and global migration. We thank all reviewers for their encouragement and support. This study has been immeasurably strengthened.
